# Evaluation of Clinical Factors Predictive of Diabetes Remission Following Bariatric Surgery

**DOI:** 10.3390/jcm10091945

**Published:** 2021-05-01

**Authors:** Isabel Mateo-Gavira, Esteban Sánchez-Toscano, Mª Ángeles Mayo-Ossorio, José Manuel Pacheco-García, Jose Arturo Prada-Oliveira, Francisco Javier Vílchez-López

**Affiliations:** 1Endocrinology and Nutrition Department, Hospital Universitario Puerta del Mar, 11009 Cádiz, Spain; isamateogavira@gmail.com (I.M.-G.); estebansanchez1994@gmail.com (E.S.-T.); 2Biomedical Research and Innovation Institute of Cádiz (INiBICA), Spain; marimayoo@gmail.com (M.Á.M.-O.); pachecadiz2@gmail.com (J.M.P.-G.); arturo.prada@uca.es (J.A.P.-O.); 3General Surgery Department, Hospital Universitario Puerta del Mar, 11009 Cádiz, Spain; 4School of Medicine, Cadiz University (UCA), 11003 Cádiz, Spain

**Keywords:** type 2 diabetes mellitus, diabetes remission, metabolic surgery, bariatric surgery

## Abstract

Bariatric surgery is an effective treatment for achieving significant weight loss and improving metabolic comorbidities such as type 2 diabetes mellitus (T2DM). The aim of our study was to investigate clinical factors related to T2DM remission in obese patients who had undergone bariatric surgery. Methods: A cohort of patients with T2DM and a minimum of class II obesity undergoing bariatric surgery had their clinical and anthropometric variables assessed. The statistical evaluation included multivariate analyses of clinical factors predicting a T2DM remission two years post-surgery. Results: 83 patients were included (mean age 44.13 ± 10.38 years). Two years post-surgery, the percentage of excess weight lost was 63.43 ± 18.59%, and T2DM was resolved in 79.5% of the patients. T2DM remission was directly related to a high body mass index (BMI) (OR: 1.886; *p* = 0.022) and the absence of macro-vascular complications (OR: 34.667; *p* = 0.002), while it was inversely associated with T2DM with a duration longer than 5 years (OR: 0.022; *p* = 0.040) and baseline insulin treatment (OR: 0.001; *p* = 0.009). 15.6% of the patients presented early complications and 20.5% developed late complications. Conclusion: In our study sample, bariatric surgery proved to be an effective and safe technique for sustained medium-term weight loss and the resolution of T2DM. A higher baseline BMI, a shorter T2DM duration, non-insulin treatment, and the absence of macro-vascular complications are factors predictive of T2DM remission.

## 1. Introduction

Type 2 diabetes mellitus (T2DM) is the metabolic comorbidity that is most associated with obesity, such that morbid obesity (BMI ≥ 40 kg/m^2^) increases the risk of developing T2DM tenfold. In patients with morbid obesity who have not yet developed T2DM, excess body adiposity is associated (in more than 90% of the cases) with insulin resistance, decreased adiponectin levels, atherogenic dyslipidemia, non-alcoholic steatohepatitis, and the secretion of pro-inflammatory cytokines—all of which entail an increased risk of acquiring cardiovascular diseases [1,2].

Bariatric surgery is the most effective therapy for reducing morbid obesity, both in terms of immediate weight loss as well as the long-term maintenance of the weight lost and the resolution of associated metabolic comorbidities. Bariatric surgery is usually reserved for patients with BMI ≥ 40 kg/m^2^ or BMI ≥ 35 kg/m^2^, but it is associated with other comorbidities (such as diabetes, hypertension, and dyslipidemia) in which dietary and pharmacological treatments have failed. In addition to their clinical status, the patients need to satisfy certain psychological conditions, which can influence the long-term outcomes of the surgery [3,4].

In the 1990s, an increase in the number of bariatric interventions aimed at reducing the BMI of patients with morbid obesity revealed that other associated comorbidities improved as well. These included T2DM via mechanisms beyond the purely anatomical ones, which were inherent in the surgical technique employed [5]. The concept of metabolic surgery was introduced by Pories et al. [6] in 1995, and various studies have followed since then. In 2004, Buchwald et al. performed a meta-analysis of 22,094 patients with obesity and T2DM who underwent different bariatric surgery procedures. The analysis indicated a 76.8% overall resolution of T2DM. Particularly noteworthy was this percentage being greater in techniques involving a malabsorption element (gastric bypass or biliopancreatic diversion) as compared to procedures involving merely a restrictive element (sleeve gastroplasty or adjustable gastric banding) [7]. Hence, some practitioners propose the use of bariatric surgery as metabolic surgery in patients with T2DM and a BMI lower than the currently accepted criteria [8].

Even though the therapeutic benefits of the different types of bariatric surgery in T2DM are clear, the factors predictive of post-surgery diabetes resolution are not so clear; data from the literature are not without controversy. We must also consider that these procedures can be irreversible and are not free of short- and long-term complications. Furthermore, although a considerable rate of improvement is obtained, not all patients achieve remission. Hence, an appropriate selection of patients is essential to optimize the outcomes.

The main objectives of the current study were to analyze the remission rates of T2DM and to identify clinical factors predictive of this remission. The study involved a sample of patients with T2DM and grade III–IV obesity undergoing bariatric surgery.

## 2. Materials and Methods

### 2.1. Study Design and Study Population

We designed a retrospective study to evaluate the possible factors involved in T2DM remission at two years post-bariatric surgery. The patients were individuals who were morbidly obese, had T2DM, and were undergoing bariatric surgery between January 2005 and December 2016. The study was approved by the Ethics Committee of our hospital (ethical approval code number: TFG-15.19).

Inclusion criteria were the following: between 18 and 60 years of age, pre-operative diagnosis of T2DM, BMI ≥ 40 kg/m^2^ or BMI ≥ 35 kg/m^2^ but with other associated metabolic comorbidities. A follow-up of the patients was conducted for at least two years post-surgery. Patients with pre-diabetes and T1DM were excluded. 

### 2.2. Surgical Procedure

The election of surgical procedure was the responsibility of the surgeon, taking into account the patient’s clinical and social characteristics. The two techniques performed were gastric bypass (a technique involving a malabsorption element) and sleeve gastrectomy (merely a restrictive intervention).

### 2.3. Evaluation of Variables Collected

The variables collated included the following: demographic data, anthropometric measurements, duration of diabetes, anti-hyperglycemic medication, micro- and macro-vascular complications, routine blood chemistry analyses (including FBG, HbA1c, total cholesterol, triglycerides, LDL-cholesterol, and HDL-cholesterol), associated metabolic comorbidities (including hypertension and dyslipidemia), and a bariatric procedure of choice together with any early and late post-operative complications.

The diagnosis of T2DM was performed according to the following American Diabetic Association (ADA) criteria: random plasma glucose ≥ 200 mg/dL (≥11.1 mmol/L) in patients with classic symptoms of hyperglycemia; fasting blood glucose (FBG) ≥ 126 mg/dL (≥6.99 mmol/L); or glycated hemoglobin (HbA1c) ≥ 6.5% [9]. Plasma glucose was determined in venous blood using the Modular DPD biochemistry system (Roche Diagnostics). HbA1c was determined in a Cobas Integra 700 analyzer (Roche Diagnostics), using an immunoturbidimetric method for anticoagulated venous whole blood. Diabetes remission was defined as FBG < 126 mg/dL (<6.99 mmol/L) and HbA1c < 6.5% in the absence of anti-hyperglycemic agents.

The chronic complications of diabetes were classified as follows: (a) macro-angiopathy—including a history of coronary heart disease, cerebrovascular disease, and/or peripheral arterial disease; (b) micro-angiopathy—including periodical evaluation and evidence of retinopathy, albuminuria and renal function, and peripheral neuropathy. Retinography was evaluated using a non-mydriatic TopCon camera. The urinary albumin excretion in 24-h urine or the spot urinary albumin-to-creatinine ratio was assessed and measured using an immunoturbidometric assay in a Cobas Integra 700 analyzer (Roche Diagnostics). The corresponding levels of albuminuria were defined as >30 mg/24 h or >30 mg/g, respectively. A distal symmetric polyneuropathy assessment was performed with a 128-Hz tuning fork and 10-g monofilament testing [10].

### 2.4. Statistical Analyses

Data were coded to ensure anonymity. Statistical analyses were performed using the SPSS program (version 15.0 for Windows, GraphPad Software, San Diego, CA, USA). Multivariate analyses were carried out using binary logistic regression models. The dependent variable was the resolution of T2DM, while the independent variables were selected based on clinical and statistical criteria of *p* < 0.05 in a bivariate analysis. Criteria for inclusion and exclusion were 0.10 and 0.15 (PIN and POUT, respectively). Likewise, variables with values of *p* < 0.300 in a bivariate analysis were manually introduced to check if the exclusion of any of these non-significant variables induced important changes in the correlation coefficients of the other clinical variables in the model, pre-set at 20% (exponent B, confounding variables).

## 3. Results

A total of 350 patients underwent bariatric surgery between January 2005 and December 2016, but only 94 were diagnosed with diabetes mellitus. Seven patients were excluded because they had not been followed up for two years, and another four were excluded due to a T1DM diagnosis. The patient-selection procedure is shown in Figure 1. Out of the 83 patients with T2DM included in the study, 45 (54.2%) underwent gastric bypass and 38 (45.8%) had sleeve gastrectomy. When analyzed according to the surgical technique performed, a higher T2DM remission rate was observed in patients who underwent gastric bypass versus sleeve gastrectomy (88.9% vs. 68.4%, respectively; *p* = 0.029). 

Demographics, clinical characteristics, anthropometric measurements, and pre-operative baseline biochemical values are shown in Table 1 and Table 2. Two years post-surgery, the percentage excess weight loss (%EWL) was 63.43 ± 18.59% and the percentage excess BMI loss (%EBMIL) was 35.14 ± 10.49%. Hypertension, dyslipidemia, and T2DM remission rates two years post-surgery were 65%, 70.2%, and 79.5%, respectively (*p* < 0.001).

Post-operative anthropometric measurements and the resolution of metabolic comorbidities are shown in Table 3. Apart from the post-operative complications caused by the surgery itself, 15.66% of patients (*n* = 13) developed early complications (mainly surgical wound infection), and 20.48% (*n* = 17) presented late complications (predominantly eventration).

Prior to conducting the bivariate analysis, the normality of distribution of quantitative variables was evaluated with the Kolmogorov–Smirnov test. Subsequent analyses used parametric or non-parametric tests. The bivariate analysis (Table 4 and Table 5) showed that patients with T2DM remission were younger (95%CI: 0.019–11.48, *p* = 0.041), had a higher baseline BMI (95%CI: −5.18–10.61, *p* = 0.025), shorter duration of T2DM (*p* = 0.006), lower HbA1c values (95%CI: 0.18–2.66, *p* = 0.047), and lower percentages of micro-angiopathy (*p* < 0.001) and macro-angiopathy (*p* < 0.001). Fewer patients presented with a pre-operative insulin requirement (*p* < 0.001), a higher percentage excess weight loss (%EWL; 95%CI: −25.91–−5.11, *p* = 0.004), and a percentage excess BMI loss (%EBMIL; 95%CI: −15.66–−4.11, *p* = 0.001) post-surgery. 

After adjusting for statistically significant variables highlighted in the bivariate analysis, as well as for clinically relevant variables (age, sex, bariatric surgery procedure, T2DM duration, BMI, presence of surgical complications, and anti-hyperglycemic medication), the multivariate analysis confirmed that a higher preoperative BMI (OR 1.886; 95%CI: 1.09–3.24; *p* = 0.022) and the absence of macro-angiopathy complications at the time of surgery (OR 34.667; 95%CI: 3.720–323.08; *p* = 0.002) were significantly associated with T2DM remission. Additionally, the clinical evolution of T2DM over the five years (OR 0.022; 95%CI: 0.001–0.846; *p* = 0.004) and the pre-surgery insulin treatment—in monotherapy (OR: 0.001; 95%CI: 0.001–0.158; *p* = 0.009) or in combination with other anti-hyperglycemic drugs (OR: 0.001; 95%CI: 0.001–0.125; *p* = 0.014)—were associated with lower odds of T2DM remission (Table 6). With respect to the surgical technique, a statistically significant difference in the remission variable was observed using a bivariate analysis. However, in multivariate analyses (after adjusting for the variables of age, sex, bariatric surgery procedure, T2DM duration, BMI, presence of surgical complications, and anti-hyperglycemic medication), the bariatric procedure did not appear as a predictor of T2DM remission.

## 4. Discussion

Achieving optimal glycemic control (defined as HbA1c < 7%) in patients with T2DM and poor metabolic control continues to be difficult despite rigorous medical treatment [11]. However, recent randomized controlled trials have provided evidence of greater efficacy of T2DM remission following bariatric surgery as compared to standard or intensive medical therapy [12,13]. As such, metabolic surgery could be considered a therapeutic alternative in patients with T2DM and a BMI lower than that currently accepted. Clinical practice guidelines do not recommend metabolic surgery routinely in patients with T2DM and a BMI between 30 and 35 kg/m^2^, considering there is not enough data on the long-term reduction of cardiovascular disease morbidity and mortality, nor on the beneficial effects on micro-vascular complications [14]. In our hospital, gastric bypass is effective in resolving metabolic comorbidities associated with obesity, and has achieved good resolution rates of hypertension, dyslipidemia, and T2DM of 71.93%, 91.38%, and 82.93%, respectively (*p* < 0.001) [15].

Although the physio-pathological mechanisms underlying the metabolic effects of bariatric surgery are complex (and continue to receive research attention), T2DM remission appears to be related to an absence of the anti-incretin effect, which is attributed to a proximal jejunum exclusion and a lower oxidation of free fatty acids, changes in gut flora, together with an early and elevated post-prandial secretion of GLP1 that occurs a few weeks post-bariatric surgery [16,17]. Recently, another important benefit of sleeve gastrectomy has been reported, i.e., a post-surgery normalization of free radicals, with values decreased to levels comparable to those of control individuals [18]. Furthermore, some authors consider bariatric surgery as an epigenetic factor influencing the anatomic and physiologic modifications induced in gene expression by these surgical techniques [19].

Our study sought to identify the factors predictive of T2DM following bariatric surgery, so as to enable the selection of patients for the surgical procedure. 

Appropriate patient selection is an important determinant of the success of bariatric surgery. It is important to know the pre-operative characteristic of the patient so as to achieve a better therapeutic response, i.e., the best outcome for the particular patient, and with a better cost-effectiveness for the institution. In our study, patients with a higher BMI, a shorter duration of clinical evolution, who are not on insulin therapy, and who had no evidence of macro- or micro-vascular complications, were more likely to have a resolution of their type 2 diabetes. The benefit to the pancreatic islet that occurs post-bariatric surgery probably depends on the beta-cell mass of the subject at the time of the intervention. Hence, if confirmed in prospective studies with larger sample sizes, a delay in selecting patients with T2DM and obesity would be avoided when considering bariatric surgery. The influence of baseline BMI on T2DM remission is debated. Some authors confirm BMI as a predictive factor in remission, while others see it as having a neutral effect [20,21,22,23]. In our study, a higher baseline (pre-operative) BMI was associated with a higher remission rate. We evaluated whether this could be influenced by the type of surgical technique employed; the rationale being that sleeve gastrectomy is frequently used in clinical practice in subjects with a lower BMI. This is also associated with lower remission rates of T2DM as compared to gastric bypass, albeit these differences do not reach statistical significance in our study. In our literature search, we were unable to identify significant relationships between pre-operative BMI and diabetes remission. Some authors have described a positive relationship between pre-operative BMI and the degree of post-surgery weight loss [24]. Other studies also show that T2DM remission is associated with weight loss post-bariatric surgery [25]. Weight loss implies a greater decrease in inflammation parameters and in insulin resistance [18]. All these data suggest that patients with a higher BMI present greater weight loss and a decrease in insulin resistance. However, randomized clinical trials with long-term clinical follow-ups are necessary to confirm these findings.

The duration of T2DM is one of the most analyzed predictive factors in diabetes literature. In our study, a T2DM duration of > 5 years was associated with an absence of remission of T2DM. Different thresholds have been proposed (between 5 and 10 years), beyond which the probability of achieving T2DM remission decreases significantly. Leoneti et al. described T2DM resolution rates close to 100% in patients with a T2DM duration of <10 years and who are undergoing sleeve gastrectomy, whereas a remission of 40% was identified in patients with a T2DM duration of >10 years [26]. Patients with a longer T2DM duration have a greater reduction in beta cell functionality. Hence, although they benefit from a reduction in insulin resistance post-surgery, they do not achieve complete T2DM remission [27]. Some authors have proposed baseline serum C-peptide as an indicator of pancreatic beta cell mass and insulin secretion, which may consequently be related to T2DM remission [28]. However, clinical relevance of C-peptide has not been well studied in diabetic literature, and cut-off points have not been clearly established.

Similarly, pre-operative insulin treatment, in monotherapy or combined with other anti-hyperglycemic medication, suggests a high level of pancreatic beta cell dysfunction that may not respond effectively to incretin secretion post-bariatric surgery [29]. This would indicate that pre-surgery insulin therapy is associated with significantly lower remission rates as compared to patients undergoing treatment with non-insulin anti-diabetic drugs [30,31]. In our study, the pre-operative insulin requirement, in monotherapy or combined with other hypoglycemic agents, was associated with an absence of T2DM remission.

In the review by Tsilingiris et al., age and sex are highlighted as factors with a neutral effect on T2DM remission in the majority of published studies [32]. Some of the authors cited in this meta-analysis detected an association between higher age and the absence of diabetes remission. This is presumably due to a greater grade of pancreatic beta cell dysfunction and, as such, with T2DM progression [28,33]. In our present study, none of these variables was identified as a factor predictive of T2DM remission. 

There is a paucity of data assessing the presence/absence of macro- and micro-angiopathy in relation to T2DM remission. In our study, we found a strong association between the absence of macro-angiopathy and T2DM remission; an aspect that certainly warrants further investigation.

It is worth noting that our study also describes the high-resolution rates of hypertension and dyslipidemia post-bariatric surgery, similar to those described in the Buchwald et al. meta-analysis [7]. Tham et al. correlated this effect with metabolic comorbidities and observed a significant reduction in cardiovascular disease risk [34]. In a long-term follow-up, the SOS study demonstrated a significant decrease in mortality from cardiovascular disease in patients who underwent bariatric surgical treatment, as compared to those who were receiving medical treatment [35].

With respect to the safety of the surgical procedures, our late complication rates are similar to those described by other authors [34]. Additionally, as in other studies, a lower incidence of complications was observed in patients undergoing laparoscopic surgery [30,36].

It would be of considerable interest to establish parameters that would influence the clinical guidelines for bariatric surgery. These include the following: (1) ideal timelines post-metabolic surgery to evaluate the resolution of T2DM; (2) the most appropriate surgical technique option (assessing risks and benefits); (3) the BMI cut-off value for surgical intervention; and (4) new pharmacological agents for T2DM treatment with proven effects on cardiovascular disease risk. Randomized studies with a long-term follow-up would be necessary to evaluate these questions.

The main strengths of our study are as follows: (1) the study was conducted in a real clinical practice setting, which included all patients with T2DM and obesity referred to our hospital over a period of more than 10 years; (2) the identification of factors have direct clinical applicability since the data could predict a better response to surgery, and consequently allow for more efficient patient selection. However, our study also has some limitations. Although the study included all the patients with T2DM and obesity treated in our center over a period of 10 years, the sample size remains relatively small (albeit comparable to other similar published studies). Nevertheless, we observed statistically significant and clinically relevant associations. The retrospective nature of the study and two years of follow-up may not be sufficient to derive definitive conclusions in the surgical treatment of T2DM because it is not known whether patients will experience T2DM recurrence in the long-term. Finally, this study was conducted in a single center; the drawback being that the results may not be generalizable to patients with different social and clinical characteristics.

## 5. Conclusions

Bariatric surgery is a safe and effective technique for sustained weight loss in the short-to-medium term. Patients with T2DM and obesity grade II (or higher) experienced an elevated remission rate of diabetes in the two-year post-bariatric surgery follow-up. A higher baseline BMI, a shorter duration of the T2DM, non-insulin treatment, and the absence of macro-vascular complications are factors predictive of T2DM remission post-surgery. Prospective studies with larger sample sizes are required to confirm the significance of clinical factors predictive of T2DM remission following bariatric surgery. The current study provides an insight into post-surgery alterations and suggests guidelines for patient selection for metabolic surgery.

## Figures and Tables

**Figure 1 jcm-10-01945-f001:**
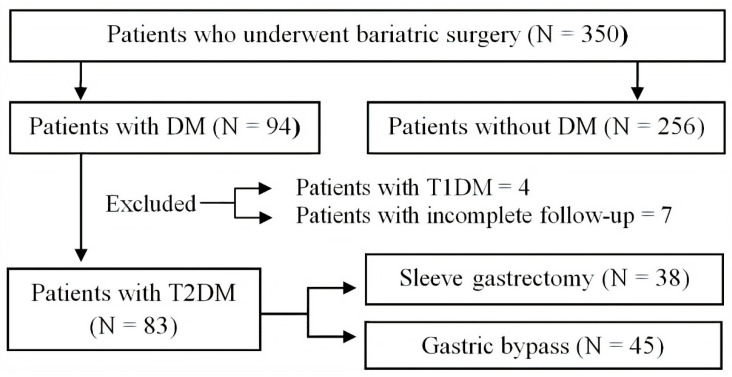
Patient-selection procedure and number of patients for each step of the process. A total of 350 patients underwent bariatric surgery between January 2005 and December 2016; only 94 of the patients were diagnosed with diabetes mellitus. Seven patients were excluded because they had not been followed up for two years, and another four due to a T1DM diagnosis. Out of the 83 patients with T2DM included in the study, 45 underwent gastric bypass and 38 had sleeve gastrectomy. DM: diabetes mellitus; T1DM: type 1 diabetes mellitus; T2DM: type 2 diabetes mellitus.

**Table 1 jcm-10-01945-t001:** Pre-operative demographics, clinical characteristics, and anthropometric measurements in patients with T2DM who underwent bariatric surgery (*n* = 83).

Variable	Value
Sex	
Female ^1^	51 (61.4%)
Male ^1^	32 (38.6%)
Age (years) ^2^	44.13 ± 10.38
Baseline weight (kg) ^2^	139.71 ± 22.27
Baseline BMI (kg/m^2^) ^3^	50.91 (46.48–54.01)
T2DM duration (years) ^3^	5.01 (3.01–15.50)
<5 years ^1^	36 (48.6%)
5–10 years ^1^	13 (17.6%)
11–20 years ^1^	20 (27%)
>20 years ^1^	5 (6.8%)
Complications of diabetes	
Micro-angiopathy ^1^	16 (23.2%)
Macro-angiopathy ^1^	7 (10.3%)
Diabetes medication	
Non-insulin treatment ^1^	61 (77.2%)
Insulin treatment ^1^	5 (6.3%)
Combination treatment ^1^	13 (16.5%)
Active tobacco habit ^1^	24 (29.3%)
Hypertension ^1^	60 (72.3%)
Dyslipidemia ^1^	57 (68.7%)

BMI = body mass index; T2DM = type 2 diabetes mellitus. ^1^ Data expressed as *n* (%). ^2^ Data expressed as mean ± standard deviation. ^3^ Data expressed as median (interquartile range).

**Table 2 jcm-10-01945-t002:** Pre-operative biochemical values of patients with T2DM who underwent bariatric surgery (*n* = 83).

Variable	Value
Total cholesterol (mmol/L) ^1^	5.16 ± 1.12
Triglycerides (mmol/L) ^2^	1.91 (1.42–2.86)
LDL–cholesterol (mmol/L) ^2^	3.27 (2.57–4.09)
HDL–cholesterol (mmol/L) ^1^	1.07 ± 0.28
Fasting blood glucose (mmol/L) ^2^	7.27 (6.05–9.16)
HbA_1c_ (mmol/L) ^1^	9.1 ± 2.2

LDL = low-density lipoprotein; HDL = high-density lipoprotein; HbA_1c_ = glycated hemoglobin. ^1^ Data expressed as mean ± standard deviation. ^2^ Data expressed as median (interquartile range).

**Table 3 jcm-10-01945-t003:** Post-operative anthropometric measurements and resolution of metabolic comorbidities.

Variable	Baseline	Two Years Post-Surgery
Weight (kg) ^1^	139.71 ± 22.27	90.31 ± 17.47
BMI (kg/m^2^) ^2^	50.91 (46.48–54.01)	33.09 ± 6.31
%EWL (%) ^1^	–	63.43 ± 18.59
%EBMIL (%) ^1^	–	35.14 ± 10.49
T2DM ^3^	83 (100)	17 (20.5)
Hypertension ^3^	60 (72.73)	21 (25.9)
Dyslipidemia ^3^	57 (68.7)	17 (21.0)

BMI = body mass index; %EWL = percentage excess weight loss; %EBMIL = percentage excess BMI loss; T2DM = type 2 diabetes mellitus. ^1^ Data expressed as mean ± standard deviation. ^2^ Data expressed as median (interquartile range). ^3^ Data expressed as *n* (%).

**Table 4 jcm-10-01945-t004:** Bivariate analysis of factors predictive of T2DM remission.

Predictive Factor	T2DM Remission (*n* = 66)	T2DM Persistence (*n* = 17)	*p* Value	95% CI
Sex				
Female ^1^	27 (40.9)	5 (29.4)	0.420	–
Male ^1^	39 (59.1)	12 (70.5)		
Age (years) ^2^	42.95 ± 10.16	48.10 ± 9.72	0.041	0.019–11.48
T2DM duration				
<5 years ^1^	34 (51.5)	2 (11.7)	0.006	–
≥5 years ^1^	24 (36.4)	14 (82.3)		
Bariatric surgery procedure				
Gastric bypass ^1^	40 (60.6)	5 (29.4)	0.029	–
Sleeve gastrectomy ^1^	26 (39.4)	12 (70.5)		
Hypertension				
Yes ^1^	47 (71.2)	13 (76.4)	0.769	–
No ^1^	19 (28.7)	4 (17.4)		
Dyslipidemia				
Yes ^1^	42 (63.6)	15 (88.3)	0.077	–
No ^1^	24 (36.3)	2 (11.7)		
Active tobacco habit				
Yes ^1^	19 (28.7)	5 (29.4)	0.917	–
No ^1^	47 (71.2)	11 (64.7)		
Micro-angiopathy				
Yes ^1^	3 (4.5)	13 (76.4)	<0.001	–
No ^1^	50 (75.7)	3 (17.4)		
Macro-angiopathy				
Yes ^1^	1 (1.51)	6 (35.3)	<0.001	–
No ^1^	52 (78.7)	9 (52.9)		
Diabetes medication				
Non-insulin treatment ^1^	56 (75.7)	5 (29.4)	<0.001	–
Insulin treatment ^1^	1 (1.5)	4 (23.5)		
Combination treatment ^1^	5 (7.5)	8 (47.1)		

T2DM = type 2 diabetes mellitus; CI = confidence interval. ^1^ Data expressed as *n* (%). ^2^ Data expressed as mean ± standard deviation.

**Table 5 jcm-10-01945-t005:** Bivariate analysis of biochemical values and anthropometric measurements.

Predictive Factor	T2DM Remission (*n* = 66)	T2DM Persistence (*n* = 17)	*p* Value	95%CI
Fasting blood glucose (mmol/L) ^1^	7.13 (5.94–8.69)	8.49 (6.44–10.32)	0.117	−14.74–46.35
HbA_1c_ (mmol/L) ^2^	10.0 ± 2.1	11.0 ± 2.2	0.047	0.18–2.66
Total cholesterol (mmol/L) ^2^	5.21 ± 1.08	4.91 ± 1.32	0.395	−38.88–15.53
Triglycerides (mmol/L) ^1^	1.83 (1.36–2.84)	2.37 (1.66–3.29)	0.187	−38.01–92.91
LDL–cholesterol (mmol/L) ^1^	3.27 (2.58–3.98)	3.26 (1.9–9.99)	0.901	−246.9–512.8
HDL–cholesterol (mmol/L) ^2^	1.06 ±0.25	1.16 ± 0.51	0.685	−17.28–24.35
Baseline weight (kg) ^2^	142.67 ± 19.83	125.30 ± 26.16	0.046	−29.39–0.41
Baseline BMI (kg/m^2^) ^2^	51.70 ± 7.88	47.79 ± 6.74	0.025	−5.18–10.6
%EWL (%) ^2^	66.17 ± 16.80	50.66 ± 21.67	0.004	−25.91–−5.11
%EBMIL (%) ^2^	36.89 ± 9.76	27.01 ± 10.24	0.001	−15.66–−4.11

T2DM = Type 2 diabetes mellitus; HbA_1c_ = glycated hemoglobin; LDL = low-density lipoprotein; HDL = high-density lipoprotein; BMI = Body mass index; %EWL = percentage excess weight loss; %EBMIL = percentage excess BMI loss; CI = confidence interval. ^1^ Data expressed as median (interquartile range). ^2^ Data expressed as mean ± standard deviation.

**Table 6 jcm-10-01945-t006:** Multivariable logistic regression analysis for factors predictive of T2DM remission.

Predictive Factor	Exp (B) (95%CI)	*p* Value
BMI baseline	1.886 (1.09–3.24)	0.022
Female sex	0.018 (0.01–1.79)	0.087
Duration of diabetes		
<5 years	3.997 (0.104–158.8)	0.456
≥5 years	0.022 (0.001–0.846)	0.040
Diabetes medication		
Insulin treatment	0.001 (0.001–0.158)	0.009
Combination treatment	0.001 (0.001–0.125)	0.014
Absence of macro-angiopathy	34.67 (3.72–323.08)	0.002

BMI = Body mass index; CI = confidence interval. Adjusted for the confounding variables of age, sex, bariatric surgery technique, T2DM duration, BMI, anti-hyperglycemic medication, and presence of T2DM chronic complications.

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
