# Peer review of "Evaluation of Clinical Factors Predictive of Diabetes Remission Following Bariatric Surgery"

_jcm, 2021, doi:10.3390/jcm10091945_

Round 1
Reviewer 1 Report
The limitation of this study is not to explain about the importance of elucidating prediction factors of improving type 2 DM by bariatric surgery. Many basic and clinical studies already showed the promising therapeutic effects of bariatric surgeries in improving type 2. Please describe why the authors planned this study with showing what is unclear for readers (in other words, what is novel and important), and what the authors aimed to show in this study in background.
I understand angiopathy and duration of DM (which may affect the diabetic chronic complications including angiopathy) can impair the therapeutic effect of bariatric surgery, but have no idea why the higher preoperative BMI contributed to the good outcome of bariatric surgery. Do you mean bariatric surgery is effective to severe obesity patients than milder ones in improving diabetes? Please discuss more detailed including basic and clinical references showing the mechanism.
Please show how to be defined the angiopathy because no explanations about this examination.
Author Response
Dear Reviewer,
Thank you for the opportunity to submit an improved version of our manuscript.
In this revised version, we have taken into account all of the comments/criticisms/suggestions. The following pages contain our itemized responses and, where necessary, we have made the appropriate changes in the text. To assist the re-evaluation process we have highlighted the changes in the text as well as in the response-to-reviewer (r-t-r) document.
As requested, we enclose:
- The modified m/s with the changes highlighted using the Track Changes function in Microsoft Word.
- The r-t-r document with the itemized responses. Please see the attachment.
Once again, we very much appreciate the extensive work of the reviewers. We hope that the revised m/s meets the quality requirements for inclusion in the JCM.
Yours sincerely,
Francisco Javier Vílchez-López, MD, PhD

Reviewer 2 Report
I was honored to review the manuscript entitled Evaluation of clinical factors predictive of diabetes remission following bariatric burgery submitted to Journal of Clinical Medicine. The study presents high quality and deals with important clinical issue, such type of study is needed. I have only few small remarks that authors should address properly.
I recommend to accept the manuscript after minor revision.
There are only some points to correct:
- please provide the list of abbreviations
- please provide the number of ethical approval
- introduction and discussion section need improvement; please provide information on how your results will translate into clinical practice
- in discussion section please provide study strong points and study limitation section
- please correct typos
All abovementioned issues are crucial for the credibility of the results. The paper can be accepted only after addressing all the issues and another subsequent review.
I recommend to accept the manuscript after minor revision.
Author Response

(The authors gave the same response as above.)

Reviewer 3 Report
The manuscript by Francisco Javier Vílchez-López et al., treats a very interesting topic about the role of bariatric surgery as a hypothetical novel therapeutic weapon. However, I have certain suggestion about the manuscript.
The manuscript is little bit confused in some parts. I suggest a review of some sentences and a review of the English to make it easier to understand and more fluent. In particular the abstract section is too confused and should be rewritten.
The methods section must be divided in different subsections and subheadings (for example: 2.1 study population; 2.2 surgical procedures; 2.3 evaluation of variables collected; 2.4 statistical analyses…)
The results into the tables should have the same units. I suggest using the same unit where it is possible or make more tables. Sometimes it is difficult to compare the data in your table.
Please try to underline, if possible, if you noted a difference in the results between sleeve gastrectomy and gastric bypass.
I invite you to enrich your Discussion and References section. In particular, I suggest you discuss about some physiopathological mechanism responsible for the diabetes remission. I recommend you read these 2 papers by Metere A et al., “The Effect of Sleeve Gastrectomy on Oxidative Stress in Obesity”. Biomedicines. 2020 Jun 19;8(6) and Metere A et al., “Factors Influencing Epigenetic Mechanisms: Is There A Role for Bariatric Surgery?” High-Throughput 2020, 9(1), 6. I think they could be helpful for your discussion and should be cited.
Author Response

(The authors gave the same response as above.)

Round 2
Reviewer 1 Report
The quality of this study is improved following reviewer's comments. The comparison between gastric bypass and sleeve gastrectomy is impressive.
Reviewer 3 Report
I appreciate the great efforts that the authors have made in response to my questions and concerns. The revision clarifies all the points.